# Bioactive Compounds of Broccoli Florets as Affected by Packing Micro-Perforations and Storage Temperature

Erika Paulsen [1], Diego A. Moreno [2], Domingo Martínez-Romero [3,*] and Cristina García-Viguera [2]

1. Facultad de Ingeniería, Instituto de Ingeniería Química, Universidad de la República, Julio Herrera y Reissig 565, Montevideo 11600, Uruguay
2. Phytochemistry and Healthy Foods Laboratory, Department of Food Science and Technology, Centro de Biología Aplicada del Sureste-Centro Superior de Investigaciones Científicas CEBAS-CSIC, Campus Universitario de Espinardo-Edificio 25, E-30100 Espinardo, Spain
3. Department of Food Technology, Escuela Politécnica Superior de Orihuela EPSO, University Miguel Hernández, Ctra. Beniel km. 3.2, 03312 Orihuela, Spain
* Correspondence: dmromero@umh.es

**Abstract:** Fresh-cut broccoli is a highly demanded product due to its convenience and high content of bioactive compounds. Unfortunately, this product shows rapid senescence and anoxia generation problems, especially when storage temperature varies. Therefore, perforation-mediated modified atmosphere packaging (PM-MAP) of broccoli florets, in different temperature scenarios, was studied. Polypropylene films with different levels of laser perforation were evaluated. After packaging, florets were stored at two temperatures: 2 °C, and 2 °C + 7 °C (during 2 d before sampling). PM-MAP slightly modified the internal composition of $O_2$ (14–20 kPa) and $CO_2$ (0.9–5 kPa) and allowed us to preserve the external quality and bioactive compounds of broccoli florets throughout 21 d, even at 7 °C. The generation of anoxia was avoided at both temperatures. PM-MAP kept broccoli mass loss below 0.5% and preserved its sensory quality. The perforation level affected evolution of firmness and glucosinolate content, especially with increasing temperature. Broccolis packaged in the film with fewer perforations showed higher firmness (0.73 ± 0.09 N/mm) and total glucosinolate content (10 ± 0.3 mg/g) compared to broccolis packaged in films with higher perforations (0.59 ± 0.05 N/mm and 8.60 ± 0.2 mg/g). Therefore, the perforation level should be taken into account in the design of packaging for fresh-cut products.

**Keywords:** perforation-mediated modified atmosphere packaging; minimally processed; storage; glucosinolates; hydroxycinnamic acids; *Brassica oleracea* var. *Italica*

## 1. Introduction

In the last years, the lifestyle of modern consumers and their desire for natural products have increased the demand for fresh-cut products, and this trend is expected to continue for the next years [1]. Fresh-cut products are fruit and vegetables that undergo operations such as washing, peeling, and cutting, to make them more convenient or ready-to-eat products, without changing their fresh-like properties [2]. As a disadvantage, the minimal processing accelerates metabolic processes and reduces the shelf life of these products compared to intact fruit and vegetable. Therefore, applying preservation technologies, such as refrigeration and modified atmosphere packaging (MAP), is crucial to ensure the quality and extend the shelf life of these commodities [3,4]. For this type of product, storage temperatures throughout the supply chain and until consumption should be below 4 °C to avoid the proliferation of microorganisms and to maintain their quality [5]. In addition, the breakage of the cold chain implies an imbalance in the internal atmosphere of the package, accelerating quality changes and the appearance of strange flavors and odors [6]. Currently, temperature maintenance can be assured at the industrial level. However, at certain points

in the cold chain, temperature increases can occur, as in the case of domestic refrigerators where temperatures are not uniform and can reach 10 °C [7].

In addition to the storage temperature, packaging technologies are of high importance to reduce the loss of fruit and vegetables. MAP has been widely used to extend the shelf life of horticultural products by controlling the surrounding gas composition of products [8]. For example, broccoli florets under MAP conditions showed a shelf life of 21 days, while florets exposed to air had a shelf life of fewer than 7 days. Moreover, it has been shown that broccoli with MAP preserves the content of bioactive compounds to a greater extent than broccoli without MAP [9]. However, MAP application is limited by the gas and water vapor permeability of commercial polymeric films, which is often low for products with high respiration rates. Therefore, approaches such as perforation-mediated MAP (PM-MAP) have been used to optimize packaging systems [10,11]. PM-MAP has proven to be a successful technology for products with high respiration rates, preventing condensation and reducing the risk of anaerobiosis [12]. Anaerobiosis generation is one of the main problems of fresh-cut products in MAP. In the case of broccoli, anaerobiosis conditions rapidly induce the production of sulfur volatile compounds which generate off-odors and reduce their shelf life [13,14]. Furthermore, anaerobic metabolism degrades dramatically the glucosinolate content of broccoli [15]. The literature reports show that PM-MAP has been applied more extensively on whole fresh fruit and vegetables than on fresh-cut products [16]. However, PM-MAP is a technique that fits better to these products, which present higher respiration rates than intact fruit and vegetables. Consequently, more studies and validations of PM-MAP are needed to develop commercially applicable solutions for minimally processed industries [16]. Responding to this need, several studies have successfully applied PM-MAP to extend the shelf life of minimally processed products [10,16–18]. For instance, Fernandez-Leon et al. (2013) [18] reported that MAP using micro-perforated polypropylene bags for fresh-cut broccoli stored at 5 °C maintained freshness during 12 d. MAP with the appropriate number of micro-perforations showed the potential to avoid excessive $CO_2$ accumulation and to prevent microbial growth in pomegranate arils during 15 d of refrigerated storage [12]. Packaging in polypropylene bags with micro-perforation extended the shelf life of broccoli florets stored at 4 °C compared to those packaged in macro-perforated films [17]. However, most of these studies are performed at storage temperatures less than 5 °C and under isothermal conditions. Nevertheless, reviewed studies show that temperature abuses often occur at all stages in the cold chain: transportation, retailed storage, retailed display of food products, or later due to mishandling of the consumer [19,20]. For fresh-cut products, temperature fluctuations between 0 °C and 10 °C in the United States and Europe, between 2 °C and 16 °C in Canada, and between 3 °C and 15 °C in Japan, have been recorded [20]. Since temperature is a crucial factor to preserve the quality of fresh fruit and vegetables, temperature fluctuations during the supply chain must be considered for the design of packaging systems for fresh-cut products [21].

The aim of this work was to evaluate the effect of three micro-perforated packaging films with different levels of micro-perforations on the quality and shelf-life of fresh-cut broccoli florets at two different temperature scenarios: 2 °C (recommended temperature); and 2 °C and subsequently transferred to 7 °C for 2 d (simulating temperature abuse or the most common domestic refrigerator temperature). The work focused on evaluating the effect of PM-MAP on the main external quality parameters (mass loss, texture, color, and overall appearance) and the main bioactive compounds of broccoli (glucosinolates and hydroxycinnamic acids).

## 2. Materials and Methods

### 2.1. Plant Material and Experimental Design

Broccoli heads (*Brassica oleracea* var. *Italica*, cv. Marathon), grown according to standard cultural practices and harvested at commercial index, were obtained from growers of Murcia, Spain (Grupo Lucas SL). Within 3 h after harvest, broccoli heads were transported to Universidad Miguel Hernández (Orihuela, Spain) (transport temperature 4 °C) and were

processed immediately. Broccoli heads, free of defects, were cut into florets with stems (3–4 cm diameter each floret). Florets were washed, sanitized in 100 mg L$^{-1}$ NaClO solution for 2 min, rinsed, centrifuged, and packaged. Florets were separated into 3 batches, 1 per each film studied. Three bioriented polypropylene (BOPP) films (30 µm thickness) with three different levels of laser perforation were evaluated. Laser perforations (100 µm) were performed in two lines along the BOPP film with separations of 15 mm (CAD15), 30 mm (CAD30), and 45 mm (CAD45). Perforated films were supplied by PDS Group SL (Murcia, Spain).

Approximately 150 ± 9 g of broccoli florets (4–5 units) were packaged in bags (26 × 20 cm). For each batch (CAD15, CAD30 and CAD45) 40 bags were confectioned. After packaging, each batch was separated into 2 groups:

- Storage at 2 °C (recommended storage).
- Storage at 2 °C and subsequently transferred to 7 °C two days before sampling (simulating temperature abuse or the most common domestic refrigerator temperature).

All samples were stored for 21 d. Figure 1 summarizes treatments performed and codes used to refer to each one.

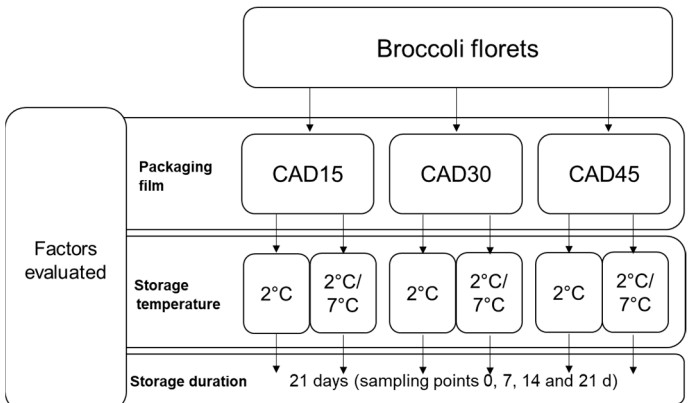

**Figure 1.** Diagram of experimental design. Packaging film: CAD15, CAD30, and CAD45 refer to the packaging films with different levels of micro-perforation (15 mm, 30 mm, and 45 mm distance between perforations, respectively). Storage temperature: 2 °C) broccoli florets were kept at 2 °C throughout all storage. The 2 °C/7 °C) broccoli florets were kept at 2 °C and transferred to 7 °C two days before sampling.

For each sampling point (0, 7, 14, and 21 d), five packages (each package constituted an experimental unit) for each condition were sampled. $O_2$ and $CO_2$ concentration inside the package, mass loss, color, texture, sensory evaluation, total glucosinolates, and hydroxycinnamic acids content were evaluated during storage.

### 2.2. Internal Atmosphere Composition

$O_2$ and $CO_2$ head space concentration were measured using a gas chromatograph (Shimadzu 14B-GC) coupled to a thermal conductivity detector, extracting a 1 mL sample directly from the package. Two measurements were made per experimental unit. Results were expressed as partial pressure (kPa) of $O_2$ and $CO_2$ in head space of packages.

### 2.3. Mass Loss

Broccoli florets were weighed prior to packaging (original mass, $m_0$) and at each sampling time ($m_t$) to determine mass loss. Mass loss (ML) was expressed as a percentage of the original mass according to the following Equation (1):

$$ML\ (\%) = (m_0 - m_t)/m_t \times 100, \tag{1}$$

*2.4. Color*

CIELAB color space parameters (L*, a*, b*, $C^*_{ab}$, and $h_{ab}$) were determined using a colorimeter CR200 (Konica Minolta, Japan). Following recommendations of Commission Internationale de L'Eclariage, 10° standard observer angles, and Standard Illuminant D65 were used. For each experimental unit, three florets were measured and within each floret three measurements were taken at different positions.

*2.5. Texture*

A compression test using a TA.XT plus Texture Analyzer (Stable Micro Systems Ltd., Godalming, UK) was used for broccoli florets' firmness quantification. Texture analyzer was equipped with a compression plate (P/100, 100 mm diameter) which applied a force that caused a deformation of 5% of broccoli floret diameter. Compression was made twice, rotating the floret 90° between measurements to obtain firmness data on two axes. Firmness values were measured as maximum force recorded divided by the displacement (N/mm).

*2.6. Sensory Evaluation*

To evaluate sensory quality of broccoli florets, overall appearance was scored using a subjective scale from 1 to 5, according to Paulsen et al. (2018) [9]. Evaluations were performed immediately after broccoli florets' removal from storage conditions. Panel members made independent evaluations per sample. Rating scale for overall appearance was: 1 = dark green, turgid, closed buds; 2 = green with yellow traces, near turgid, minor defects; 3 = light green, trace limp, moderate defects; 4 = yellow, limp, major defects; and 5 = very yellow, very limp, open buds. A score of 3 was considered as the limit of marketability and a score of 4 was the limit of edibility [9].

*2.7. Glucosinolate (GSL) and Hydroxycinnamic Acids (HCAs) Content*
2.7.1. Sample Extraction

Sample extraction was carried out according to Baenas et al. (2016) [22] with modifications according to Paulsen et al. (2021) [23]. Freeze-dried samples were extracted with methanol (70% *v/v*), heated at 70 °C for 30 min and centrifuged (15,000× *g*, 15 min). Supernatants were collected and filtered through a 0.22 μm Millex-HV13 filter (Millipore, Billerica, MA, USA).

2.7.2. HPLC-DAD Analysis of GSL and HCAs

The extracts were analyzed and quantified in Waters HPLC-DAD system (Waters Chromatography SA, Barcelona, Spain). Intact GSL and HCAs were identified following UV spectra and order of elution according to Paulsen et al. (2021) [23]. For quantitation of GSL and HCAs, chromatograms were registered at 227 and 330 nm, respectively. GSL were quantified using sinigrin and glucobrassicin (Phytoplan, Germany) as external standards of aliphatic and indole glucosinolates, respectively. HCAs were quantified using chlorogenic and sinapinic acid as external standards. Results were expressed as mg per gram of dry weight (mg/g) for GSL and as μg per gram of dry weight (μg/g) for HCAs. These measurements were performed in triplicate for each condition.

*2.8. Data Analysis*

An analysis of variance (ANOVA) was performed for all quality parameters evaluated considering the level of perforation of film, storage temperature, storage time, and their interaction as variation factors. When effects were significant, honestly significant differences were calculated using Tukey's test. Differences were considered significant when $p < 0.05$. Statistical analysis was performed using R version 4.1.0 (R Core Team, Vienna, Austria, 2021).

## 3. Results and Discussion

### 3.1. Internal Atmosphere Composition

Adjustment of gaseous composition surrounding fresh-cut fruit and vegetables is one of the main factors to preserve their quality and extend their shelf life. The $O_2$ reduction and $CO_2$ increase in MAP and PM-MAP are derived from the respiration rate of the product and the gas transmission rate of the packaging film [10]. In the case of PM-MAP, the $O_2$ and $CO_2$ transmission rate depends mainly on the perforated area of the film [24]. In the present study, all micro-perforated packages showed a significant change in internal $O_2$ and $CO_2$ composition (with respect to air composition) during storage (Figure 2). The level of perforation and storage temperature had a significant impact on these variables ($p < 0.001$). For broccoli florets kept at 2 °C, the $CO_2$ concentration showed significant differences between films. $CO_2$ levels were $1.01 \pm 0.04$, $1.83 \pm 0.03$, and $2.69 \pm 0.08$ kPa (at day 7 of storage) for CAD15, CAD30, and CAD45, respectively. The difference in $CO_2$ composition between films was maintained throughout storage ($0.78 \pm 0.03$, $1.55 \pm 0.06$, and $2.30 \pm 0.13$ kPa for CAD15, CAD30, and CAD45, respectively, at day 21). Regarding $O_2$ concentration, CAD15 and CAD30 did not show significant differences ($19.3 \pm 0.32$ kPa on average) at day 7 of storage, and CAD45 presented a significantly lower $O_2$ concentration ($17.9 \pm 0.39$ kPa). At day 21, CAD30 and CAD45 did not present significant differences ($19.1 \pm 0.33$ kPa on average), while CAD15 showed the highest $O_2$ concentration ($20.4 \pm 0.49$ kPa).

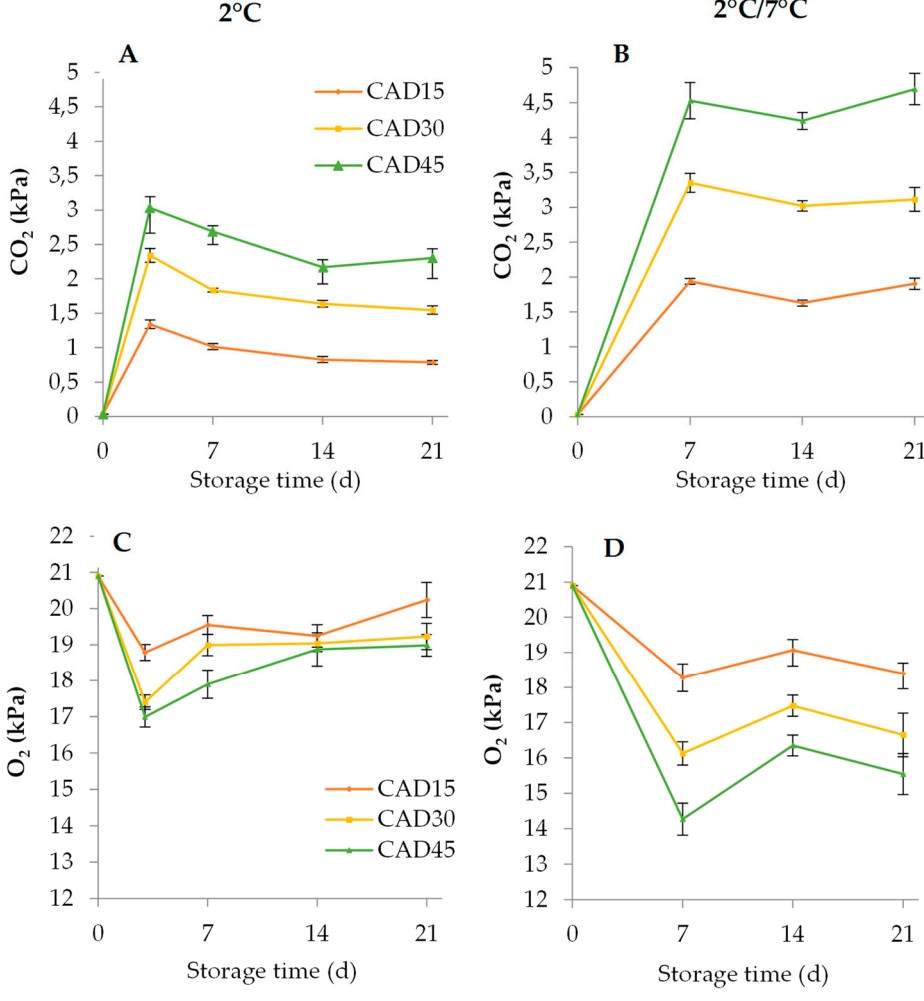

**Figure 2.** Evolution of internal atmosphere composition ($O_2$ and $CO_2$) of broccoli florets packaged in films with different level of micro-perforation (CAD15, CAD30 and CAD45) throughout storage at 2 °C (**A**,**C**) and 2 °C/7 °C (**B**,**D**). Vertical bars represent standard error (n = 5, n is the number of samples analyzed).

The increase in storage temperature significantly decreased $O_2$ and increased $CO_2$ concentration inside packages compared to those that were kept at 2 °C ($p < 0.01$). This behavior is mainly due to the effect of temperature on the respiration rate of the vegetable [25]. The difference in gas composition between films was more marked in this storage condition (2/7 °C), where $O_2$ and $CO_2$ composition between films were significantly different throughout all the storage period ($p < 0.001$) (Figure 2). At this condition, CAD15 samples showed higher levels of $O_2$ and lower of $CO_2$ (18.3 ± 0.40 kPa and 1.9 ± 0.04 kPa, respectively, at day 7), and CAD45 samples showed lower $O_2$ and higher $CO_2$ concentration (14.3 ± 0.45 kPa and 4.5 ± 0.26 kPa, respectively, at day 7). Florets packaged in CAD30 presented intermediate levels of $O_2$ and $CO_2$ (16.1 ± 0.33 kPa and 3.3 ± 0.13 kPa, respectively, at day 7). Similar gas composition evolutions were reported by Lucera et al. (2011) [26] for broccoli florets packaged in polypropylene micro-perforated films, who observed a slow decrease in $O_2$ at the beginning of the storage (5 °C) and an equilibrium value of about 16 kPa.

The risk of anaerobiosis is one of the main problems presented by MAP technology in products with high respiration rates, especially during temperature fluctuations. Anaerobiosis generation induces anaerobic metabolism, which is undesirable due to off-odor generation and the risk of anaerobic pathogenic microorganism proliferation [9]. The results showed that PM-MAP technology was useful in avoiding the risk of anaerobiosis against small temperature variations (from 2 °C to 7 °C) (in no case was the concentration of $O_2$ less than 14 kPa and the $CO_2$ higher than 5 kPa). Therefore, these conditions could be directly applied to fresh-cut broccoli marketing. However, for its application in supply chains with poor temperature control (temperatures above 7–8 °C), more studies are needed to ensure that micro-perforated films prevent the generation of anoxia and generate adequate levels of $O_2$ and $CO_2$.

### 3.2. Mass Loss

It is well known that the dehydration of broccoli florets is one of the leading causes that shorten their shelf life [9]. As expected, mass loss increased during storage ($p < 0.05$). However, the mass loss of florets was below 0.5 % of initial mass during the entire storage period, regardless of film and storage temperature (Table 1). This is in agreement with Fernández-León et al. (2013) and Caleb et al. (2016), who reported weight losses of less than 1% for broccoli florets packaged in a modified atmosphere [18,27]. The increase in storage temperature (from 2 °C to 7 °C) did not have a significant impact on mass loss. However, broccolis transferred to 7 °C tended to have greater mass loss compared to those held at 2 °C. Films with different levels of perforation also had no significant effect on the mass loss of broccoli florets, although CAD45 and CAD15 tended to have the lowest and highest mass loss throughout storage, respectively. Our results agree with Lucera et al. (2011) [26], who reported a weight loss of less than 3% for broccoli florets stored for 24 d at 5 °C, regardless of the number of film micro-holes. In all conditions of this study, mass loss was less than the maximum limit of marketability (6%) and was not a factor limiting the shelf-life of broccoli florets [21]. Therefore, PM-MAP technology was a solution to preserve this important quality attribute, avoiding senescence symptoms of broccoli florets at least for 21 d.

### 3.3. Color

Color is one of the main outer quality parameters of broccoli florets which limited their shelf life. If the appropriate conditions are not selected for its storage and distribution, broccoli flower buds undergo rapid yellowing and loss of green color [26]. In this study, neither the storage temperature nor the level of perforation of films had a significant effect on the colorimetric parameters of broccoli florets ($p > 0.05$). For all samples, L*, $C_{ab}$, and $h_{ab}$ remained unchanged during storage (Table 2). Figure 3 shows that there was no appreciable color change under any of the conditions assayed. Changes in broccoli color parameters under MAP depend on the gas composition inside the package [28]. However, when the storage temperature is kept within the recommended values (<4 °C), the effect of the

atmosphere is usually not significant [9]. This could explain why no impact of $O_2$ and $CO_2$ levels on broccoli color were detected. The increase at 7 °C for 2 days had no significant effect on color parameters. Several authors have reported that broccoli retains its color at concentrations of 1–10 kPa for $O_2$ and 5–10 kPa for $CO_2$ [17,29]. The application of MAP (using conventional continuous films) reaches these gaseous compositions and inhibits surface color loss [9,18]. Generally, in PM-MAP these strict gaseous conditions are not achieved due to the greater permeability of micro-perforated films. This study proves that is not necessary for such low levels of $O_2$ and such high levels of $CO_2$ to maintain the color of broccoli florets during refrigerated storage (2 °C). Therefore, PM-MAP demonstrates to be an effective technology to preserve fresh broccoli color.

**Table 1.** Mass loss, external appearance, and hydroxycinnamic acids content (HCAs) evolution of broccoli florets packaged in films with different levels of micro-perforation (CAD15, CAD30 and CAD45) throughout storage at 2 °C and 2 °C/7 °C. Data are expressed as means ± standard error. Same letter in the same column means no significant differences according to Tukey's test ($p < 0.05$).

| Storage Temperature | Packaging Condition | Day | Mass Loss (%) (†) | | External Appearance (1–5) (†) | | HCAs (µg/g) (‡) | |
|---|---|---|---|---|---|---|---|---|
| 2 °C | CAD15 | 0 | 0.0 ± 0.0 | a | 1.0 ± 0.0 | a | 316 ± 26 | efg |
| | | 7 | 0.15 ± 0.04 | abc | 1.3 ± 0.3 | ab | 405 ± 42 | bcdefg |
| | | 14 | 0.30 ± 0.03 | bcdef | 1.7 ± 0.3 | efg | 422 ± 19 | abcde |
| | | 21 | 0.36 ± 0.02 | efg | 1.7 ± 0.3 | efg | 525 ± 15 | ab |
| | CAD30 | 0 | 0.0 ± 0.0 | a | 1.0 ± 0.0 | a | 316 ± 26 | efg |
| | | 7 | 0.08 ± 0.02 | ab | 1.6 ± 0.3 | abc | 508 ± 16 | ab |
| | | 14 | 0.18 ± 0.04 | bcdef | 1.5 ± 0.2 | bcde | 308 ± 28 | efg |
| | | 21 | 0.28 ± 0.04 | efg | 2.0 ± 0.4 | gh | 465 ± 14 | abcd |
| | CAD45 | 0 | 0.0 ± 0.0 | a | 1.0 ± 0.0 | a | 316 ± 26 | efg |
| | | 7 | 0.08 ± 0.01 | ab | 1.3 ± 0.3 | abc | 419 ± 28 | abcdef |
| | | 14 | 0.16 ± 0.06 | bcde | 1.4 ± 0.2 | bcd | 279 ± 26 | g |
| | | 21 | 0.26 ± 0.01 | defg | 1.9 ± 0.2 | fgh | 522 ± 35 | ab |
| 2 °C/7 °C | CAD15 | 0 | 0.0 ± 0.0 | a | 1.0 ± 0.0 | a | 316 ± 26 | efg |
| | | 7 | 0.18 ± 0.06 | bcdef | 1.6 ± 0.3 | bcdef | 356 ± 15 | cdefg |
| | | 14 | 0.26 ± 0.03 | defg | 1.7 ± 0.2 | defg | 351 ± 25 | defg |
| | | 21 | 0.44 ± 0.09 | h | 2.4 ± 0.5 | h | 380 ± 14 | abcd |
| | CAD30 | 0 | 0.0 ± 0.0 | a | 1.0 ± 0.0 | a | 316 ± 26 | efg |
| | | 7 | 0.12 ± 0.02 | abcd | 1.5 ± 0.3 | bcde | 437 ± 17 | abcde |
| | | 14 | 0.23 ± 0.02 | cdefg | 1.7 ± 0.2 | defg | 372 ± 24 | cdefg |
| | | 21 | 0.33 ± 0.04 | gh | 2.2 ± 0.4 | hi | 540 ± 32 | a |
| | CAD45 | 0 | 0.0 ± 0.0 | a | 1.0 ± 0.0 | a | 316 ± 26 | efg |
| | | 7 | 0.09 ± 0.01 | abc | 1.7 ± 0.3 | defg | 483 ± 15 | abc |
| | | 14 | 0.18 ± 0.01 | bcdefg | 1.6 ± 0.2 | cdefg | 289 ± 10 | fg |
| | | 21 | 0.31 ± 0.03 | fgh | 2.1 ± 0.4 | hi | 422 ± 15 | abcde |

(†) n = 5, (‡) n = 3.

**Table 2.** Colorimetric parameters (L*, $C_{ab}$ and $h_{ab}$) of broccoli florets packaged in films with different levels of micro-perforation (CAD15, CAD30 and CAD45) and storage for 21 days at 2 °C and 2 °C/7 °C, and comparison with florets freshly harvest (day 0). Data are expressed as means ± standard error. Same letter in the same column means no significant differences according to Tukey's test ($p < 0.05$).

| Day | Storage Temperature | Packaging Condition | L* | | $C_{ab}$ | | $h_{ab}$ | |
|---|---|---|---|---|---|---|---|---|
| 0 | | | 37.87 ± 0.45 | a | 12.23 ± 0.44 | a | 133.1 ± 0.4 | a |
| 21 | 2 °C | CAD15 | 38.58 ± 0.43 | a | 12.69 ± 0.51 | a | 132.0 ± 0.6 | a |
| | | CAD30 | 39.73 ± 0.30 | a | 12.23 ± 0.47 | a | 132.5 ± 0.8 | a |
| | | CAD45 | 39.02 ± 0.36 | a | 13.98 ± 0.68 | a | 131.3 ± 0.6 | a |
| | 2 °C/7 °C | CAD15 | 39.75 ± 0.22 | a | 13.9 ± 0.60 | a | 131.0 ± 0.6 | a |
| | | CAD30 | 39.31 ± 0.41 | a | 12.51 ± 0.50 | a | 133.8 ± 0.6 | a |
| | | CAD45 | 39.57 ± 0.27 | a | 14.49 ± 0.66 | a | 132.3 ± 0.8 | a |

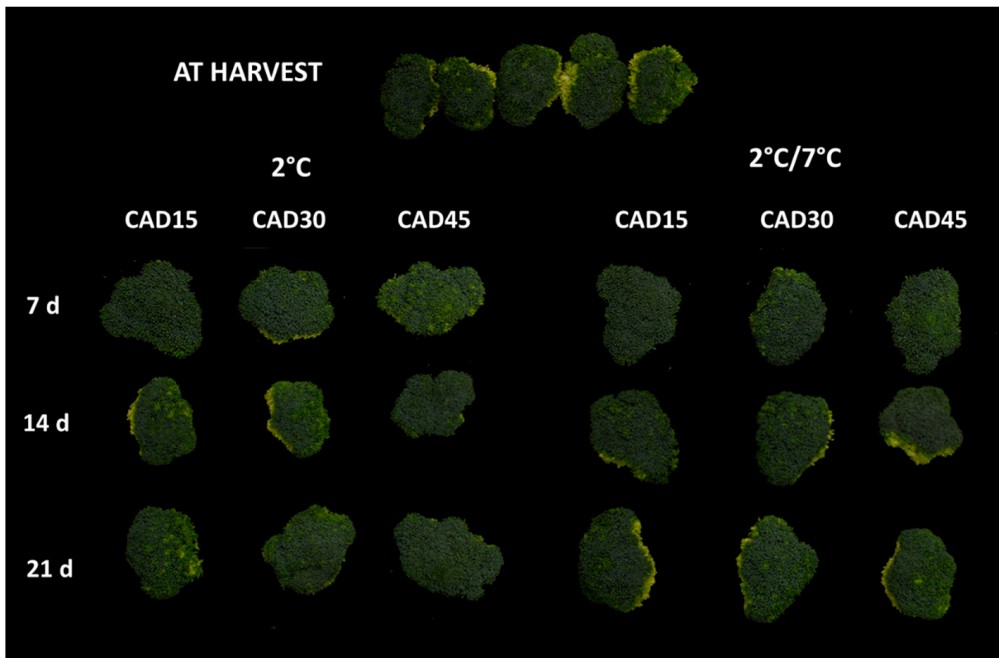

**Figure 3.** Evolution of visual quality of broccoli florets packaged in films with different levels of micro-perforations (CAD15, CAD30 and CAD45) and stored at two conditions of temperature (2 °C and 2 °C/7 °C) during 21 d.

*3.4. Texture*

The texture test carried out aims to assess the compactness (firmness to pressure) of broccoli florets. This is an aspect to highlight since in most studies a hedonic evaluation is performed to assess this texture parameter [30].

Regardless of packaging film and storage temperature, broccoli florets showed a decrease in firmness values throughout storage ($p < 0.0001$) (Figure 4). No significant differences were observed in firmness evolution between storage temperature conditions. Therefore, for broccoli florets under PM-MAP, 2 days at 7 °C did not have an impact on firmness compared to holding the product at 2 °C ($p = 0.7236$). Significant differences in firmness were found depending on the level of perforation of the film ($p = 0.0004$). Broccolis packaged in CAD15 showed a rapid decline of firmness in the first week of storage, and then remained without significant changes until the end of storage. At day 7, CAD30 and CAD45 samples showed significantly higher firmness compared to CAD15. On days 14 and 21 of storage, CAD45 samples showed the highest firmness values, and florets packaged in CAD15 and CAD30 did not show significant differences in firmness between them (for both storage temperatures). The effect of MAP in maintaining vegetable firmness is usually related to the control of mass loss, and several works have found a high correlation between mass loss and firmness loss of plant tissues [31–33]. Although no significant differences were found in mass loss between films, broccoli packed in CAD15 tended to present the highest mass losses and CAD45 the lowest, both at 2 °C and 2/7 °C. Therefore, this could explain firmness evolution. Furthermore, the differences in firmness could be attributed to the in-package gas composition. Broccoli that retained the most firmness (CAD45) were those exposed to lower $O_2$ and higher $CO_2$ concentrations (see Section 3.1). This agrees with reports showing that the activity of enzymes involved in the biochemical processes leading to deterioration in the cell structure is reduced when levels of $O_2$ are reduced and $CO_2$ levels rise [33]. It is important to note that the difference in firmness between florets packaged in different films did not have an impact on their external quality, since no significant differences were found in overall appearance (see Section 3.5). These results can be corroborated by Figure 3, where all samples show closed and tightly crowded together individual buds, with a firm appearance. Results indicate that the atmosphere modification

and mass retention generated by the PM-MAP application allowed the preservation of broccoli florets' firmness during refrigerated storage. The film with the lowest level of perforation (CAD45) was the one that best preserved the firmness of the product.

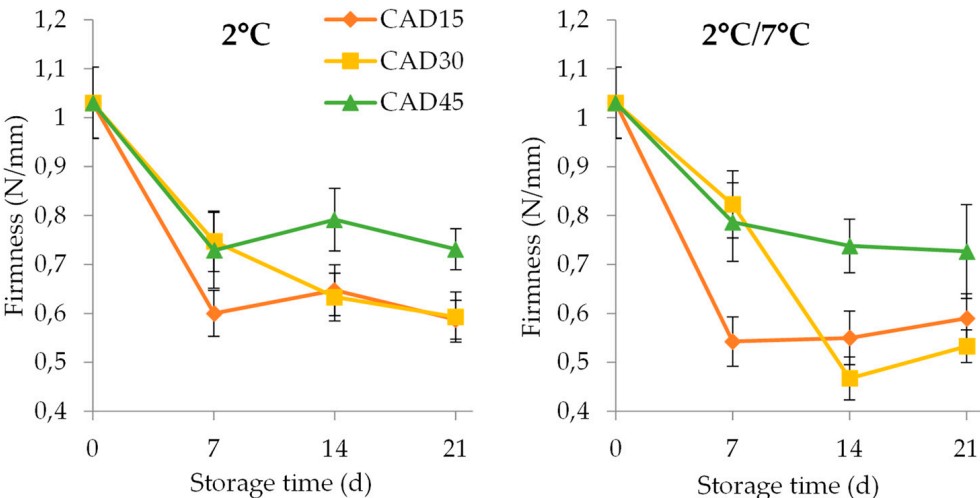

**Figure 4.** Firmness evolution of broccoli florets packaged in films with different levels of micro-perforation (CAD15, CAD30 and CAD45) throughout storage at 2 °C and 2 °C/7 °C. Vertical bars represent standard error (n = 5).

### 3.5. External Appearance

In all conditions, broccoli florets showed an increase in their overall appearance score during storage, indicating a decline in their sensory quality (Table 1) ($p < 0.001$). However, all samples maintained their values of overall appearance under the threshold score throughout all storage periods. Films with different levels of perforation did not have a significant impact on overall appearance evolution ($p = 0.3087$). For broccoli florets packaged in CAD30 and CAD45, no significant differences were found between storage temperatures. A significant effect of temperature on overall appearance was found in broccoli florets packaged in CAD15, 1.7 ± 0.3 and 2.4 ± 0.5 for broccolis storage at 2 °C and 2 °C/7 °C, respectively. Thus, films with lower perforation levels (CAD30 and CAD45) helped to mitigate the negative effect of increased temperature on the organoleptic quality of broccoli florets. Figure 3 shows photos of the external appearance of broccoli florets where it can be seen that it did not change significantly throughout storage. Florets stored for 21 days look like freshly harvested florets, regardless of film and temperature conditions. Therefore, PM-MAP technology successfully contributed to preserving the sensory quality of broccoli florets.

### 3.6. Total Glucosinolate Content

Two chemical classes of glucosinolates were detected, one aliphatic: glucoraphanin (GRA) and three indolic: glucobrassicin (GB), neoglucobrassicin (NGB), and 4-methoxygluc-obrassicin (MGB). The four compounds were identified in all samples analyzed, regardless of days of storage, storage temperature, and packaging film applied. Similar results were reported by Vallejo et al. (2003) [34], who studied the effect of refrigerated transport on glucosinolate content in broccoli heads of Marathon cv. and found GRA, GB, NGB, and MGB in all samples during every time period evaluated.

Regarding the glucosinolate profile of the broccoli cultivar used in this experiment (Marathon), the predominant glucosinolate was GRA accounting for 41.6% of total glucosinolates, followed by NGB, GB, and MGB with 32.7%, 19.5%, and 6.2%, respectively. Although the predominant compound was from the chemical group of aliphatic glucosinolates, the predominant group was the indolic one with 58% of total glucosinolates. The glucosinolate profile coincides with several studies that report GRA as the predominant

glucosinolate, and NGB and GB as the predominant glucosinolates within the group of indolics [17,35,36]. No significant changes were observed in the glucosinolate profile during refrigerated storage in any of the packaging conditions studied.

In all samples, an increase in total glucosinolate content was observed in the first week of storage (Figure 5A,B). This can be explained by the fact that minimal processing leads to tissue mechanical damage, which could activate the defense mechanisms of the vegetable increasing the synthesis of glucosinolates during the first days of storage [18]. No significant differences were found between broccolis packaged in different films during storage at 2 °C, where glucosinolate content was maintained until the end of storage (Figure 5A). However, a significant effect of packaging film was observed in broccoli florets transferred to 7 °C for 2 d ($p = 0.0012$) (Figure 5B). Broccolis packaged in CAD45 showed higher total glucosinolate content during the entire storage period compared to broccolis packaged in CAD15 and CAD30, which did not show differences between them. At the end of storage, CAD15 and CAD30 samples showed the same glucosinolate content as fresh broccoli, and CAD45 samples had a significantly higher content. Therefore, there was no effect of packaging film on glucosinolate content while the storage temperature was kept low (2 °C). However, packaging film had a significant effect on these compounds during temperature abuse (when storage temperature increased from 2 °C to 7 °C). Several studies have reported the impact of storage temperature in the glucosinolate content of broccoli heads and broccoli florets [9,17]. High temperatures cause higher losses during postharvest storage. The effect of packaging film on glucosinolates with increasing storage temperature is in line with that reported by Paulsen et al. (2018) [9], who conclude that modification of the atmosphere helps to mitigate the effects of high storage temperatures on glucosinolate content. In addition, PM-MAP also had an impact on firmness, and glucosinolate content could be correlated with this quality parameter. When tissue integrity is lost, myrosinase enters into contact with glucosinolates and the rate of degradation of these compounds is increased [37]. Thus, the lower firmness loss of CAD45 samples implies a lower loss of compartmentalization and this could result in less degradation of glucosinolates.

In conclusion, PM-MAP combined with refrigeration showed to preserve the total glucosinolate content of broccoli florets during 21 d of storage. This is a finding to be highlighted because until now conservation of glucosinolates has been reported for broccoli florets under MAP, where the modification of in-packaged atmosphere is stronger (about 1–5 kPa for $O_2$ and 10–13 kPa for $CO_2$) than PM-MAP [9,17,37]. The low concentrations of $O_2$ and high concentrations of $CO_2$ reported as beneficial to preserve glucosinolates of broccoli florets are risky. Temperature fluctuations during the supply chain and the high respiration rate of broccoli florets tend to generate anoxia conditions, which leads to the detriment of the product [9]. This study identified conditions of refrigerated storage (2 °C and 2 °C/7 °C) and moderated concentrations of $O_2$ and $CO_2$ (14–20 kPa and 1–5 kPa, respectively), that conserve the content of the main bioactive compounds of broccoli florets and that could avoid the risk of anoxia.

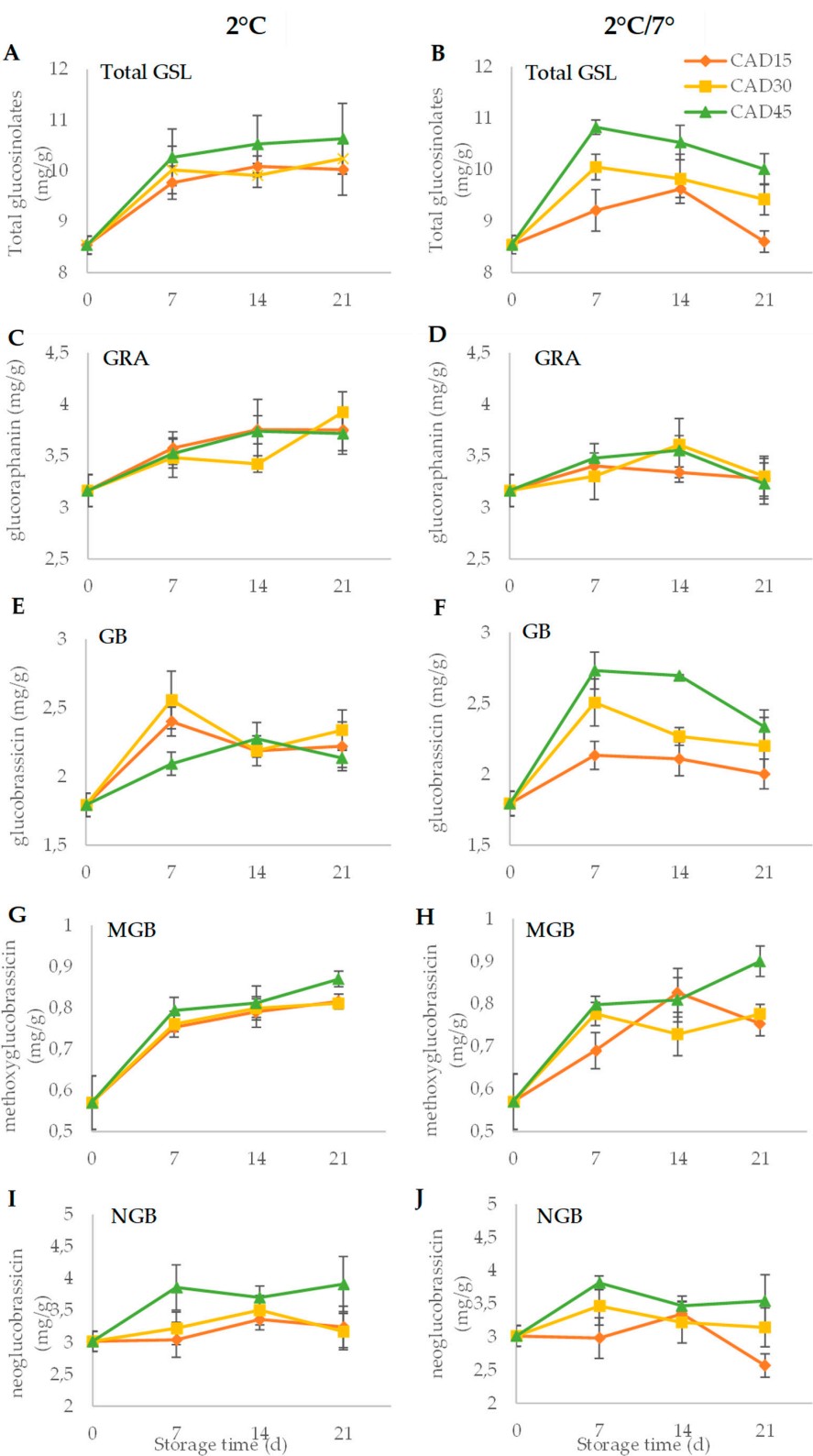

**Figure 5.** Evolution of total glucosinolate content (**A**,**B**), GRA (**C**,**D**), GB (**E**,**F**), MGB (**G**,**H**), and NGB (**I**,**J**) of broccoli florets packaged in films with different levels of micro-perforation (CAD15, CAD30 and CAD45) stored at 2 °C and 2 °C/7 °C. Vertical bars represent standard error (n = 3).

### 3.7. Individual Glucosinolate Content

All the conditions preserved the GRA content of broccoli florets (Figure 5C,D). In the case of samples kept at 2 °C, GRA showed a tendency to increase during storage, regardless

of packaging film (Figure 5C). This is in accordance with Winkler et al. (2007) [24], who found stable levels of GRA in broccoli heads (Marathon cv.) wrapped in polyethylene after storage for 3 days at 8 °C with pre-storage at 1–4 °C. In addition, similar results were found by Rybarczyk-Plonska et al. (2016) [35], who reported that broccoli heads (Marathon cv.) pre-stored for 7 days at 0 °C and then stored for 3 days at 10 °C had higher GRA content than at harvest. In these studies, the atmospheric composition is not reported. However, studies on broccoli florets under MAP (1–10 kPa $O_2$ and 10–20 kPa $CO_2$) found that at the beginning of storage, GRA remains constant, but a significant loss is observed at the end of storage (at 4–8 °C for 12–23 days) [9,17,28]. Consequently, maintaining GRA content requires careful consideration of packing film and storage temperature (among other considerations such as cultivar selection, agronomic conditions, etc.). Packaging films and storage temperatures applied in this study allowed for preserving GRA, the compound precursor of sulforaphane, which is highly recognized for its beneficial effects in neurodegenerative and cardiovascular diseases.

GB showed a similar evolution of total glucosinolate content (increased in the first week and then remained constant or decreased to initial values) (Figure 5E,F). The effect of packaging film was observed in broccolis transferred to 7 °C. In this condition, GB content was 2.13 ± 0.01, 2.50 ± 0.16, and 2.73 ± 0.13 mg/g for broccolis packaged in CAD15, CAD30, and CAD45, respectively (day 7). CAD45 showed higher content of GB compared to CAD15 during all storage periods (at 2 °C/7 °C). MGB content increased in all conditions, regardless of film packaging (Figure 5G,H). At the end of storage, the MGB content of broccoli packaged in CAD45 was significantly higher (0.87 ± 0.03 mg/g and 0.90 ± 0.04 mg/g for 2 °C and 2 °C/7 °C, respectively), with respect to florets packaged in CAD15 (0.81 ± 0.02 mg/g and 0.73 ± 0.03 mg/g) and CAD30 (0.81 ± 0.01 and 0.78 ± 0.02 mg/g). For all conditions, NGB content remained constant throughout storage. Despite no significant impact found between broccolis packaged in different films, samples CAD45 and CAD15 tended to have the highest and lowest NGB content, respectively (Figure 5I,J). Several authors have reported an increase in indole glucosinolate content [7]. Recently, it has been reported that the expression of genes associated with the biosynthesis of indolic glucosinolates increased during postharvest storage [38]. Moreover, protein degradation increases during senescence and free amino acids are accumulated, which are precursors of the biosynthesis of glucosinolates. These phenomena could be the cause of the increase in indolic glucosinolate. These processes are enzyme mediated and depend on vegetal metabolism; consequently, storage temperature and $O_2$ and $CO_2$ concentration would have an impact on indolic glucosinolate content, as observed in our results.

In conclusion, the results indicate that refrigerated storage combined with slight modification of the atmosphere (PM-MAP) was effective in preserving individual and total glucosinolate content. In addition, packaging film had an effect on these compounds (CAD45 showed higher glucosinolate content), especially when storage temperature increased (7 °C).

*3.8. Hydroxycinnamic Acids (HCAs)*

The study of HCAs in foods has gained importance in recent years due to their potential health benefits. In addition to its antioxidant capacities, some reports attribute antidiabetic effects and inhibitory efficacy against breast and hematologic malignancies [38].

Sinapic acids and their derivatives were the predominant group of HCAs identified in broccoli florets. The evolution of these compounds over time was the same for all the conditions assayed (Table 1), observing an increase during storage. Our results coincide with the evolution of HCAs reported by Paulsen et al. (2022) [36] in broccoli heads stored at 2 °C in different packaging films. There are few studies that report the evolution of HCAs content during postharvest storage. However, several studies reported an increase in the antioxidant capacity of broccoli, which could be correlated with HCAs' activity and in agreement with the HACs increase observed in this experiment [39,40]. The synthesis of HCAs can be induced by the generation of reactive oxygen species during the senescence

process, which could explain its increase during storage [41]. Statistical analysis did not show differences in HCAs content between broccoli florets in different packaging films or storage temperatures ($p > 0.05$). In conclusion, regardless of the level of film perforation, PM-MAP preserved and even increased, the HCA content of broccoli florets during refrigerated storage (2 °C) and during temperature abuse (2 °C/7 °C).

## 4. Conclusions

Results show the suitability of PM-MAP as a feasible low-cost technology to extend the shelf-life of fresh-cut broccoli florets.

This study identified moderate concentrations of $O_2$ (14–20 kPa) and $CO_2$ (0.9–5 kPa), that combined with refrigerated storage (2 °C) conserved bioactive compounds and outer quality of broccoli florets throughout 21 days, even during 2 days at 7 °C. In addition, PM-MAP prevented the generation of anoxia, the main problem during the commercialization of this product. The film perforation level affected the firmness and glucosinolate content of florets, especially in the face of increases in storage temperature. Broccoli florets packaged in the film with fewer perforations (CAD45) showed higher firmness value and glucosinolate content. Therefore, the selection of the perforation numbers is a relevant aspect to be taken into account in the packaging design for fresh-cut products.

The information generated in this study is easily transferable to minimally processed industries and could be used as input for the design of new packaging for fresh vegetables.

**Author Contributions:** Conceptualization, D.M.-R.; methodology, D.M.-R.; software, E.P.; validation, D.A.M. and C.G.-V.; formal analysis, E.P.; investigation, E.P.; resources, D.M.-R.; data curation, E.P.; writing—original draft preparation, E.P.; writing—review and editing, D.M-R.; visualization, E.P.; supervision, D.A.M., D.M.-R., and C.G.-V.; project administration, D.M.-R.; funding acquisition, D.M.-R. All authors have read and agreed to the published version of the manuscript.

**Funding:** The authors are grateful to ANII (Agencia Nacional de Innovación e Investigación) for conceding Erika Paulsen an internship at Phytochemistry and Healthy Foods Laboratory of the Department of Food Science and Technology at CEBAS-CSIC (scholarship reference: POS_EMHE_2018_1_10-07740). This research has been supported by the University Miguel Hernández (UMH) through Proyect PAR3265/19.

**Institutional Review Board Statement:** Not applicable.

**Informed Consent Statement:** Not applicable.

**Data Availability Statement:** All data generated or analyzed during this study are included in this published article.

**Acknowledgments:** The authors would like to thank Grupo Lucas (Murcia, Spain) and PDS GROUP (Murcia, Spain) for the contribution of broccoli heads and packaging films, respectively.

**Conflicts of Interest:** The authors declare no conflict of interest.

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
