# Peer review of "Bioactive Compounds of Broccoli Florets as Affected by Packing Micro-Perforations and Storage Temperature"

_coatings, doi:10.3390/coatings13030568_

Round 1
Reviewer 1 Report
The manuscript titled "Bioactive compounds and outer quality of broccoli florets as affected by the micro-perforations of packaging and small variations in storage temperature" is well written with good scientific value. The following points can be addressed before further processing of the manuscript.
1. Title of the manuscript is very long. The number of words can be reduced to convey the meaning with the use of less number of words.
2. The abstract can also include significant results regarding the presence of various compounds.
3. Kindly provide the full form of the words at their first usage. For instance, the word CAD could not be found in the manuscript
4. Error bar is missing for CAD30 and CAD 45 plots in figure 2A. If the error bars are so small, kindly reduce the size of the bullet shape.
5. Conclusion can be rewritten by including all the salient experimental results.
6. Using only recent references (preferable references from the last 5 years) is recommended for better understanding.
Author Response
We thank his most valuable comments and questions. We have carefully revised this manuscript following the comments received.
Point 1: Title of the manuscript is very long. The number of words can be reduced to convey the meaning with the use of less number of words.
We agree with the reviewer’s assessment. We changed the title to be more concise. Title now reads: “Bioactive compounds of broccoli florets as affected by packing micro-perforations and storage temperature”
Point 2: The abstract can also include significant results regarding the presence of various compounds
We have tried to improve the abstract by addressing the reviewer's comment (see lines 18-27).
Point 3: Kindly provide the full form of the words at their first usage. For instance, the word CAD could not be found in the manuscript
CAD is not an abbreviation or acronym. CADn is the nomenclature used to refer to the different films (CAD15, CAD30, CAD45), and it refers to the “cadence” of laser perforation (distance between two perforations) along the perforation line.
Point 4: Error bar is missing for CAD30 and CAD 45 plots in figure 2A. If the error bars are so small, kindly reduce the size of the bullet shape.
We reduce the size of the bullet shape so that the error bars are visible (see Figure 2).
Point 5: Conclusion can be rewritten by including all the salient experimental results.
Thank you for the comment. The conclusions were rewritten including the most relevant results (see lines 477-489).
Point 6: Using only recent references (preferable references from the last 5 years) is recommended for better understanding.
Old references that were not strictly necessary have been eliminated.

Reviewer 2 Report
Please explain which is the control sample
Line 91 and elsewhere in the text: I suggest changing the term “normal” from “normal domestic refrigerator temperature” phrase, not to indicate that 7°C is the indicated temperature, while using a lower temperature would be “not normal”/ not indicated.
As my understanding is that the authors refer here to most common temperature used in domestic refrigerators I suggest changing throughout the text from “normal… temperature” to “most common/ used” or similar other rephrasing as authors decide.
Line 99 -100 temperature during transport should be indicated.
Figure 1 suggestion to change ”storage time” into “storage duration”
Line 224: risk of pathogenic microorganism proliferationà Most food pathogens are aerobic due to aerobic storage of food, so I suggest adding “anaerobic pathogens”
3.3 Color Missing table with L, a, b, C, h values
3.5 Please correct the title “Glucosinltale profile” and 3.6. Indivudual glucosinolate
Line 469 I suggest indicating the exact temperature of 7°C after “temperature abuse”
Author Response
We thank his most valuable comments and questions. We have carefully revised this manuscript following the comments received.
Point 1: Please explain which is the control sample
In all the experiences carried out by the group, broccoli florets packaged in macro-perforated films are used as a control. In this packaging condition, florets stay exposed to air composition (the gas transmission rate of the film is high) and rapidly lose their mass and characteristic green color. This behavior has been widely reported in the literature for broccoli exposed to air (Fernández-León, Fernández-León, Lozano, Ayuso, & González-Gómez, 2013; Fernández-León, Fernández-León, Lozano, Ayuso, Amodio, et al., 2013; Jia et al., 2009; Lucera et al., 2011; Schouten et al., 2009). Therefore, this information was not included. Additionally, it does not provide relevant information to evaluate the effect of micro-perforations on quality, which is the objective of this work.
Point 2: Line 91 and elsewhere in the text: I suggest changing the term “normal” from “normal domestic refrigerator temperature” phrase, not to indicate that 7°C is the indicated temperature, while using a lower temperature would be “not normal”/ not indicated. As my understanding is that the authors refer here to most common temperature used in domestic refrigerators I suggest changing throughout the text from “normal… temperature” to “most common/ used” or similar other rephrasing as authors decide.
Thanks for the suggestion. We changed “normal domestic refrigeration temperature” to “the most common domestic refrigeration temperature” (see lines 90 and 113).
Point 3: Line 99 -100 temperature during transport should be indicated.
We indicated the temperature during transport from growers to our lab (see line 113).
Point 4: Figure 1 suggestion to change “storage time” into “storage duration”.
We took the suggestion and changed “storage time” to “storage duration” (see Figure 1).
Point 5: Line 224: risk of pathogenic microorganism proliferationà Most food pathogens are aerobic due to aerobic storage of food, so I suggest adding “anaerobic pathogens”
“anaerobic pathogens” was added (see line 223).
Point 6: 3.3 Color Missing table with L, a, b, C, h values
We did not include a table because the colorimetric parameters remain unchanged during storage, regardless of the film and temperature applied. However, according to the reviewer's comment, now we include a summary table (Table 2, line 284) with L, Cab, and hab values. The table is attached below.
Table 2. Colorimetric parameters (L*, Cab, and hab) of broccoli florets packaged in films with different levels of micro-perforation (CAD15, CAD30, CAD45) and storage for 21 days at 2°C and 2°C/7 °C, and comparison with florets freshly harvest (day 0). Data are expressed as means ± standard error. Same letter in the same column means no significantly differences according to Tukey’s test (p < 0.05).
|
Day |
Storage temperature |
Packaging condition |
L* |
Cab |
hab |
|
||||
|
0 |
|
|
37,87 ± 0,45 |
a |
12,23 ± 0,44 |
a |
133,1 ± 0,4 |
a |
||
|
21 |
2°C |
CAD15 |
38,58 ± 0,43 |
a |
12,69 ± 0,51 |
a |
132,0 ± 0,6 |
a |
||
|
CAD30 |
39,73 ± 0,30 |
a |
12,23 ± 0,47 |
a |
132,5 ± 0,8 |
a |
||||
|
CAD45 |
39,02 ± 0,36 |
a |
13,98 ± 0,68 |
a |
131,3 ± 0,6 |
a |
||||
|
2°C/7°C |
CAD15 |
39,75 ± 0,22 |
a |
13,9 ± 0,60 |
a |
131,0 ± 0,6 |
a |
|||
|
CAD30 |
39,31 ± 0,41 |
a |
12,51 ± 0,50 |
a |
133,8 ± 0,6 |
a |
||||
|
CAD45 |
39,57 ± 0,27 |
a |
14,49 ± 0,66 |
a |
132,3 ± 0,8 |
a |
||||
Point 6: 3.5 Please correct the title “Glucosinltale profile” and 3.6. Indivudual glucosinolate
We are not sure we understood this comment. We changed the titles to “3.5. Total glucosinolate content” and “3.6. Individual glucosinolate content” (see lines 352 and 408).
Point 7: Line 469 I suggest indicating the exact temperature of 7°C after “temperature abuse”
We agree and have corrected it (see line 481).

Reviewer 3 Report
This is a manuscript review for the article ''Bioactive compounds and outer quality of broccoli florets as affected by the micro-perforations of packaging and small variations in storage temperature''
The manuscript is well-written and the results are presented clearly.
The results and discussion were combined which gave little room for comparison of results with work done on a similar food item. Although the behaviour of vegetables will be similar in certain conditions, it is important to attempt a comparison of like for like. A few more comparisons to recent work like that of reference 30 will be beneficial.
Author Response
We thank his most valuable comments and questions. We have carefully revised this manuscript following the comments received.
The manuscript is well-written and the results are presented clearly.
Point 1: The results and discussion were combined which gave little room for comparison of results with work done on a similar food item. Although the behaviour of vegetables will be similar in certain conditions, it is important to attempt a comparison of like for like. A few more comparisons to recent work like that of reference 30 will be beneficial.
Comparisons with results from other studies were included, as suggested by the reviewer (see lines: 241-243, 248-250, 275-276, 464-466). Reference to results of other works had already been made in the first version of the manuscript. Please see lines: 216-219, 356-359, 387,411-419, where reference has already been made to the results of other studies.
